# Non-Linear Spectral Unmixing for the Estimation of the Distribution of Graphene Oxide Deposition on 3D Printed Composites

**Giorgio Licciardi [1],\*, Costantino Del Gaudio [2] and Jocelyn Chanussot [1]**

1   GIPSA-Lab, Grenoble Institute of Technology, 38000 Grenoble, France; jocelyn.chanussot@grenoble-inp.fr
2   Fondazione E. Amaldi, 00133 Roma, Italy; costantino.delgaudio@fondazioneamaldi.it
\*   Correspondence: giorgio-antonino.licciardi@gipsa-lab.grenoble-inp.fr

**Abstract:** Hyperspectral analysis is a well-established technique that can be suitably implemented in several application fields, including materials science. This approach allows us to deal with data samples containing spatial and spectral information at very high resolution, thus enabling us to evaluate materials properties at a nanoscale level. As a proof of concept, hyperspectral imaging was here considered to investigate 3D printed polymer matrix composites, considering graphene oxide (GO) as a nanofiller. Commercial polycaprolactone and polylactic acid filaments were firstly treated with GO to be then printed into testing specimens. Raman analysis was performed to assess the GO distribution on samples surface by mapping different regions of interest and the collected data were the input of a custom-made algorithm for hyperspectral image analysis, tailored to detect the GO signature. Findings showed a valuable matching to Raman maps and were also characterized by the positive feature of avoiding to set specific conditions to perform the investigation as GO Raman distribution was carried out by fixing the wavenumber at 1580 cm$^{-1}$, which is representative of the G band of the nanofiller. This occurrence might lead to an uneven intensity representation related to possible peak shifts which can bias the acquired results. Differently, hyperspectral imaging needs a minimal set of data input, i.e., the spectral signatures of neat materials, to directly identify the searched nanomaterial. More in-depth investigations need to be performed to fully validate the proposed approach, but the here presented results already show the potential and versatility of hyperspectral analysis in the materials science field.

**Keywords:** hyperspectral imaging analysis; graphene oxide; 3D printing

## 1. Introduction

Composite fabrication can be regarded as a real option to deal with novel materials with properly designed properties for specific technological applications. This outcome can be achieved when each step of the production process is carefully controlled, starting from an effective nanofiller dispersion within the matrix, as the expected performance is strictly dependent on the fine interaction of the parent materials. A detailed characterization is thus a crucial issue to be addressed to evaluate the properties of the final component in detail and to tune the processing variables, to optimize the desired output. In general, traditional approaches to identify materials are based on the comparison of detected Raman spectra with reference spectral libraries. However, the identification of more than one material in a sample, and their characterization is still an open issue.

In this regard, non-linear spectral analysis is an alternative and valuable approach to elaborate Raman maps collected from 3D printed composites including graphene oxide (GO) [1], which was selected as a model nanomaterial to test the reliability of the here proposed method. GO is a single

layer of sp2 bonded carbon atoms arranged in a hexagonal lattice modified with functional groups, e.g., epoxides, alcohols, and carboxylic acids [2], which strongly improve the hydrophilic characteristics and contribute to forming stable aqueous dispersions.

In Raman spectroscopy, the signal recorded for every single pixel is the result of the interaction of light with several materials at the surface. In this framework, the term spectral unmixing refers to the identification of the so-called endmembers, the spectral responses of the single constituent materials, and to the determination of the amount of information with which each endmember contributes to the detected signal [3]. According to recent literature, the spectral mixture model can be classified as a linear or nonlinear process [4]. When the incident light interacts with only one material and the scattered signals from different materials are mixed within the sensor, we can refer to the linear spectral mixing model [5–7]. On the other hand, when the light scattered from one material interact with other materials, having different chemical or physical characteristics, before reaching the sensor, we can refer to the nonlinear mixing model. Referring to Raman spectroscopy, the incident monochromatic light is scattered by molecules and most of the scattered light has the same frequency of the incident one (Rayleigh scattering), but some fraction has different frequency due to the interaction between light characteristics and molecular vibration (Raman scattering). Because the resolution of the spectrometer sensor covers a surface larger than a single molecule, the incident light interacts with more than one molecule before being detected by the detector.

When the radiation interacts with a composite of two or more materials that are intimately mixed (such as sand grains), we can refer to a microscopic nonlinear mixture. In this kind of mixtures, the interaction between light and materials requires extremely complex modeling, taking into account also the non-Lambertian properties of the materials. This explains why most of the literature is dedicated to linear approaches with very few papers presenting only approximations of nonlinear models [8,9]. Alternative solutions have been proposed using machine learning, in particular with neural networks [10–12]. In this kind of approach, it is possible to train a neural network to learn linear and nonlinear correlations in the data. However, to detect all the possible correlations, it will be necessary to build a very large training dataset covering all the possible scenarios given the problem under investigation [13]. In practical applications, this ideal training dataset is not always available, thus, it is necessary to find alternative solutions. An effective solution is based on the use of neural networks to project the original data into a feature space through the use of nonlinear transformations. This results in the linearization of the data projected in the feature space, thus allowing the effective use of linear unmixing approaches. In particular, in this paper, we propose an approach that uses Nonlinear Principal Component Analysis (NLPCA) to project the Raman data into a linear feature space [14], and a classical linear unmixing approach applied directly to the linearized features to detect endmembers and estimate their abundances.

The here presented study is aimed to highlight the potential of an alternative investigational technique for materials characterization. As a case study, commercial polycaprolactone and polylactic acid filaments were modified with GO to 3D print testing composites, to implement Raman findings for the spectral unmixing analysis. Raman mapping is generally carried out by fixing a specific wavenumber representative of the material to be detected; however, this might lead to an uneven intensity representation related to possible peak shifts which can bias the acquired results. Differently, hyperspectral imaging needs a minimal set of input data, i.e., the spectral signatures of the neat materials, to directly identify the searched nanomaterial, which allows one to limit the number of the experimental constraints to be set and thus perform a reliable material assessment.

## 2. Theoretical Background

According to the linear mixture model definition, each pixel vector in the original map can be modeled using the following expression:

$$X(i,j) = \sum_{z=1}^{P} \Phi_z(i,j) \cdot E_z + n(i,j) \tag{1}$$

where $X(i,j)$ represent the pixel vector at discrete coordinates $(i,j)$, $E_z$ denotes the spectral signatures of the endmembers, $\Phi_z(i,j)$ is a scalar value indicating the fractional abundance of the endmember $z$ in the pixel vector, and $n(i,j)$ is a noise vector. When dealing with linear mixing models, it is possible to define two physical constraints, i.e., the sum-to-one constraint:

$$\sum_{z=1}^{P} \Phi_z(i,j) = 1 \tag{2}$$

and the non-negativity constraint:

$$\Phi_z(i,j) \geq 0 \tag{3}$$

To find the optimal solution of the fully constrained linear spectral mixture problem, the following requirements are to be satisfied:

- Identification of the correct number of the endmembers
- Determination of a set of endmembers and their corresponding abundances fractions at each pixel.

Effective solutions have been proposed so far to deal with the first requirement [15]. However, the second requirement can be addressed in two steps: endmember extraction and abundance estimation. In the last decades, several algorithms have been developed for the extraction of spectral endmembers, such as the pixel purity index (PPI) [16], N-FINDR, and vertex component analysis (VCA) [17,18]. Once identified the endmembers, the abundance estimation consists of solving a constrained optimization problem, minimizing the residual between the observed spectral vectors and the linear space spanned by the inferred spectral signatures, under the constraints of non-negativity and/or the sum to one.

In general, linear mixing dominates macroscopic interactions, when the incident light interacts with only one material before reaching the sensor. In the case of Raman spectroscopy, the light is scattered by multiple materials and the interactions consist of photons emitted by molecules of one material that are absorbed by molecules of another material, which may in turn emits more photons.

Assuming then a nonlinear mixture model, in Raman spectroscopy maps, each pixel vector can be modeled as

$$X(i,j) = f(E_z, \Phi_z(i,j)) + n(i,j) \tag{4}$$

where f is an unknown nonlinear function that defines the interaction between $E_z$ and $\Phi_z(i,j)$. Deriving a physically based approach to nonlinear unmixing is an extremely complex task that requires identifying parameters describing all the possible interactions between light and materials that are very hard or impossible to obtain. To avoid complex physical models, neural networks are generally considered, thanks to their ability to approximate complex functions and the high potential in decomposing nonlinear mixed pixels [12]. Following a classical supervised approach, to train an effective neural network, it is important to know how many endmembers are present in the data and their spectral signature to design an optimal unmixing function f. However, in many practical cases, it is not possible to know in advance this information, thus the unmixing function may result incomplete.

Whether linear or nonlinear, the general framework of spectral unmixing can be summarized in dimensionality reduction, endmember identification, and abundance estimation, as depicted in Figure 1.

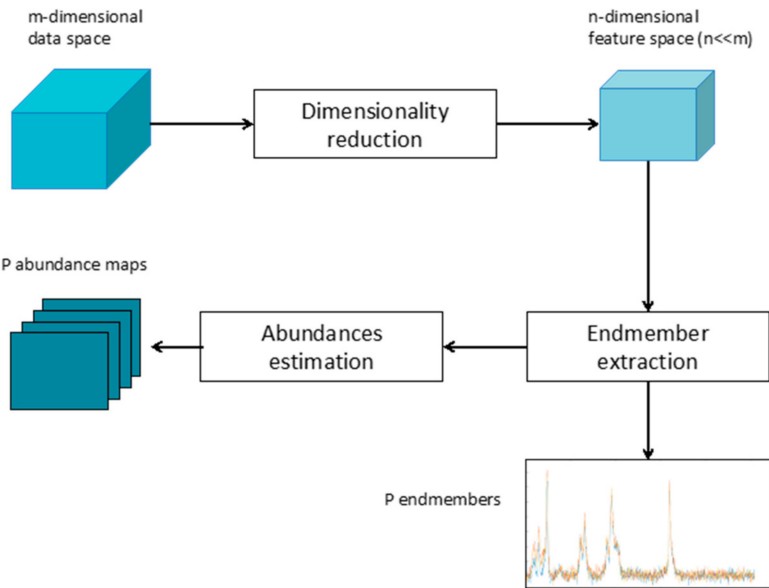

**Figure 1.** Sketch of the general spectral unmixing workflow.

The dimensionality reduction is an optional processing step and its application depends on the dimension of the original data. However, in the case of Raman images, characterized by thousands of highly correlated bands, the use of a dimensionality reduction approach can be beneficial, improving computation time, complexity, and performances of the whole unmixing process [19]. Many dimensionality approaches have been proposed in the literature, from feature selection to feature extraction. A comparative analysis of the effectiveness of different methods can be found in reference [20].

Several linear approaches have been proposed so far to extract the endmembers from a given data. These approaches can be divided into geometrical techniques that consider the mixed vector enclosed within a simplex statistical framework that determines endmembers through parameters estimation, and methods that model the linear mixture as a sparse regression problem. Given the data, the abundance of each identified endmember can be quantified by solving a constrained optimization problem, which minimizes the residual between each of the observed spectral vectors and the space spanned by the inferred endmembers [21]. It is worth noting that using linear approaches in images presenting a nonlinear mixture may not only detect all the endmembers present in the scene but also may result in the identification of non-existing endmembers.

Taking into account that modeling all the possible nonlinear interactions between light and materials characterizing Raman spectroscopy is an extremely complex task, the use of NLPCA [22] is here proposed to linearize the original data into a linearized feature space while reducing the original data dimensionality. This will then allow the implementation of linear unmixing models for the extraction of endmembers and abundance estimation. In particular, the N-FINDR and SUnSAL algorithms as described in [23,24] will be considered for the endmember extraction and the solution of the constrained least square problem, respectively.

## 3. Materials and Methods

### 3.1. Nonlinear Principal Component Analysis

NLPCA was here implemented to project the original data into a lower dimensionality linear feature space. The proposed approach is based on the use of Auto-Associative Neural Network (AANN) or auto-encoder. Firstly, introduced in the 1980s, the autoencoders are now widely used as one of the most powerful deep learning approaches. An autoencoder is a conventional feed-forward neural network having a symmetrical topology with the input and output layers having the same

number of nodes. According to the requirements of the NLPCA, the autoencoder was designed to have three hidden layers, with the central one, defined as the bottleneck layer, having a smaller dimension than the other hidden layers [25]. During the training phase, the nodes of the bottleneck layer force the other hidden layers, defined as coding/decoding layers, to compress the information by removing correlations, noise, and redundancies. Once trained, the information extracted from the bottleneck layer represents the lower dimensionality feature space. It is worth noting that the number of nodes in the bottleneck layer defines the dimension of the target feature subspace, while the nodes in the coding and decoding layers are related to the complexity of the mapping and demapping functions to and from the feature space. However, the selection of these values cannot be defined a priori. Thus, to identify the optimal neural network topology, it is possible to apply a simple heuristic grid search algorithm varying recursively the number of nodes of the hidden layers and evaluating the value of the Means Square Error (MSE) error [26]. Then, the topology presenting the smallest error is selected. A detailed description of the NLPCA implementation can be found in reference [22].

*3.2. Endmember Extraction and Abundance Estimation*

Several approaches have been developed for the automatic or semiautomatic extraction of spectral endmembers, among which N-FINDR is one of the most powerful. N-FINDR is a geometric algorithm that identifies the simplex of maximum volume that can be inscribed within the data set [23]. The initialization of the algorithm is carried out by selecting a random set of q endmembers $\{E_1, E_2, \ldots, E_q\}$, with $q \leq n + 1$, and n corresponding to the dimension of the feature space. The volume of the simplex is then defined by

$$V\left(E_1, E_2, \ldots, E_q\right) = \frac{\left|\det \begin{bmatrix} 1, 1, & \ldots, 1 \\ E_1, E_2, \ldots, E_q \end{bmatrix}\right|}{(q-1)!} \tag{5}$$

To identify the maximum volume of the simplex, this value is recalculated by testing each pixel vector X(i,j) in the first endmembers position:

$$\begin{aligned} V\left(X(1,1), E_2, \ldots, E_q\right) \\ V\left(X(1,2), E_2, \ldots, E_q\right) \\ \ldots \\ V\left(X(r,c), E_2, \ldots, E_q\right) \end{aligned} \tag{6}$$

where *r and c represent the* number of rows and columns of the map. When one of the volumes calculated in Equation (6) is greater than the volume obtained with the original endmembers set, the evaluated endmember is replaced with the pixel corresponding to the maximum volume, resulting in a new set of endmembers. This procedure is then iterated by testing the volumes in the other endmember's positions, retaining the combinations corresponding to the maximum volumes. At the end of this process, the resulting volume should represent the maximum volume for the specific data. It is important to note that according to our approach, the obtained endmembers are representative of the feature space, thus to obtain their equivalent in the spectral domain, it will be necessary to reproject them back into the original data space.

The estimation of the fractional abundances of each endmember within each pixel in the image can be obtained by minimizing the total squared error, under the constraints of non-negativity and/or the sum to one

$$\begin{aligned} \min_x \left(\tfrac{1}{2}\right) \|Ea - S\|_2^2 \\ x \geq 0 \\ 1^T x = 1 \end{aligned} \tag{7}$$

where $E \in \mathfrak{R}^q$ refers to the matrix containing the q endmembers, $a \in \mathfrak{R}^k$ represents the fractional abundance vector, and $S \in \mathfrak{R}^k$ is the observed mixed pixel. The solution of the optimization problems

is then achieved through the SUnSAL algorithm that is based on the Constrained Split Augmented Lagrangian Shrinkage Algorithm (C-SALSA) methodology to effectively solve a large number of constrained least-squares problems sharing the same matrix system in reference [24].

### 3.3. Specimens Fabrication and Raman Analysis

The proposed approach has been tested on Raman maps acquired from 3D printed polymer-based composites. The graphene oxide (GO; powder, oxidation: 4–10%) was supplied by Sigma-Aldrich (Milan, Italy), while bidistilled water was purchased by Carlo Erba Reagenti (Milan, Italy). Polycaprolactone (PCL) and polylactic acid (PLA) filaments were supplied from 3D4Makers (Haarlem, The Netherlands) and Formfutura BV (Nijmegen, The Netherlands), respectively. All materials were used as received.

Specimens were fabricated by preliminary depositing GO on the surface of PCL and PLA filaments. For this aim, GO was suspended in bidistilled water (0.5% w/v) and ultrasonicated for 30 min. Filament samples were then soaked in the resulting suspension, magnetically stirred for 3 h, dried at 40 °C once recovered, and subsequently loaded in the N2 FFF 3D printer (Raise 3D Inc., Irvine, CA, USA). The geometric models, representative of the composites to be printed, were processed by the ideaMaker software (Raise 3D Inc., Irvine, CA, USA) to be sliced in the Z direction. For the 3D printing process, the temperature of the nozzle (0.4 mm diameter) was set at 160 °C and that of the build platform at 20 °C for PCL samples, while the temperature of the nozzle was set at 205 °C and that of the build platform at 40 °C for PLA samples. In both cases, they were fabricated by alternatively stacking layers in a regular pattern in the 0°/90° directions, the sample size being $8 \times 8 \times 3.2$ mm$^3$.

Raman characterization was performed to evaluate the GO presence and distribution on the specimen surface using an InVia Raman microscope (Renishaw, UK). The analysis was carried out on GO powder, 3D printed neat samples and 3D printed PCL/PLA-GO samples. Carbon nanofiller distribution on specimen surface was assessed by collecting Raman maps, randomly located. A fixed area of $250 \times 120$ μm (step $5 \times 5$ μm) was investigated using a laser with a wavelength of 532 nm, 1 s exposure time, and 2 accumulations for each acquired spectrum. Collected spectra were then minimally processed to correct the baseline and remove the possible presence of cosmic rays (WiRE software, Renishaw, UK).

## 4. Results and Discussion

Figure 2 shows the 3D printed structures here considered for the prosed unmixing analysis. The average size of the deposited strands was $352.0 \pm 5.8$ μm for PCL-GO and $409.6 \pm 5.3$ μm for PLA-GO, respectively; further data were previously reported in reference [1].

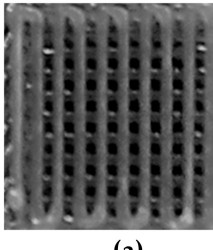 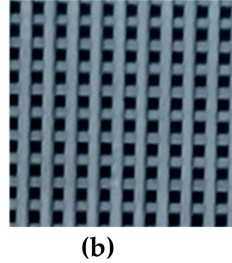

**(a)** **(b)**

**Figure 2.** Pictures of polycaprolactone-graphene oxide (**a**) and polylactic acid-graphene oxide (**b**) 3D printed structures.

Figure 3 shows the Raman spectra of GO powder, neat PCL, PCL-GO, PLA, and PLA-GO 3D samples. GO spectrum is characterized by the two typical peaks at about 1320 cm$^{-1}$ (D band) and 1580 cm$^{-1}$ (G band), which is clearly identifiable in both spectra acquired from the printed composites.

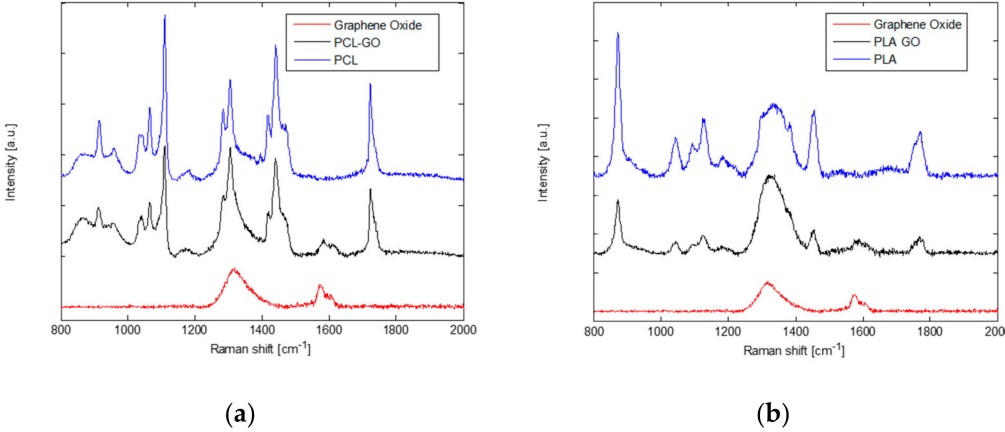

(**a**)  (**b**)

**Figure 3.** Raman spectra of neat polycaprolactone and polycaprolactone-graphene oxide (**a**), and neat polylactic acid and polylactic acid-graphene oxide (**b**). In both panels the Raman spectrum of graphene oxide is reported for comparison, the G band is clearly detectable in the two composites.

GO distribution was investigated by scanning a fixed area, randomly selected, on different locations on PCL- and PLA-based composites. Figure 4 shows representative maps considering the intensity peak at 1580 cm$^{-1}$.

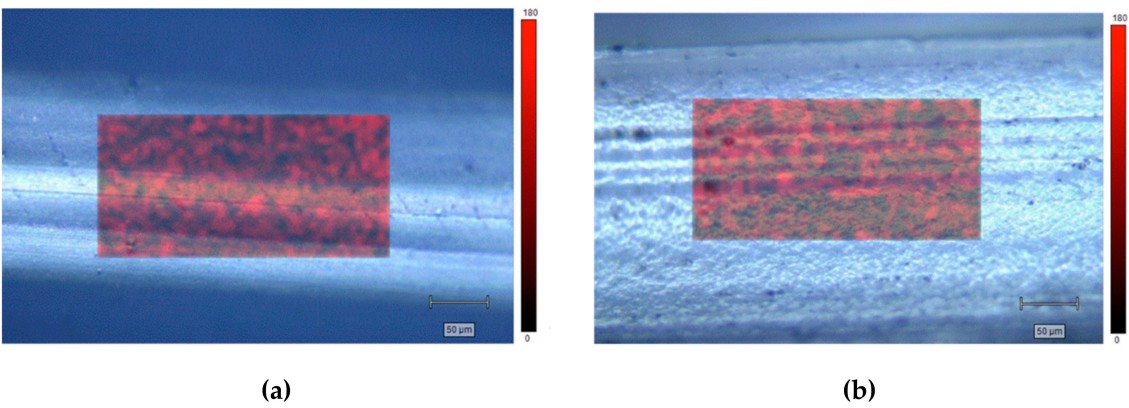

(**a**)  (**b**)

**Figure 4.** Raman maps of PCL-GO (**a**) and PLA-GO (**b**).

For each dataset (PCL-GO and PLA-GO), two different approaches have been carried out for the selection of the endmembers. In the first case, the endmembers have been selected manually from a spectral library, while in the second one the endmembers have been derived from data using the N-FINDR endmember extraction technique. The spectral library considered in this study is composed of a series of Raman measurements of pure materials. In particular, we selected three different measurements of GO powder, and three different measurements of PCL and PLA filaments, respectively.

Two different metrics have been used to compare the performance of the proposed technique depending on the endmember extraction approach. Regarding the endmember extracted from the data, the assessment has been carried out using the Spectral Angle Distance (SAD) between each extracted endmember and the set of available GO spectral signatures e and PCL or PLA. The SAD measures the spectral angle formed by n-dimensional vectors and is defined as follows:

$$\text{SAD}(X(i,j), X(r,s)) = \cos^{-1} \frac{X(i,j) \cdot X(r,s)}{\|X(i,j)\| \|X(r,s)\|} \tag{8}$$

where low SAM values mean high spectral similarity between the compared signatures.

A second metric for the evaluation of the unmixing algorithms is based on the assumption that a set of high-quality abundance maps and their corresponding endmembers may allow the reconstruction

of the original data with higher precision than a set of low-quality endmembers/abundance maps. Specifically, the metric used is the Root Mean Squared Error (RMSE) between the original and the reconstructed data, which can be defined as follows:

$$\text{RMSE}\left(I^{(O)}, I^{(R)}\right) = \frac{1}{s \times l} \sum_{i=1}^{s} \sum_{j=1}^{l} \left( \frac{1}{n} \sum_{k=1}^{n} \left[ x_k^{(O)}(i,j) - x_k^{(R)}(i,j) \right] \right)^{1/2} \tag{9}$$

where $I^{(O)}$ is the original map, and $I^{(R)}$ is a reconstructed version of $I^{(O)}$, obtained using Equation (1) with the set of endmembers used for the unmixing process and their corresponding estimated fractional abundances.

## 4.1. PCL-GO Dataset

The obtained map of the PCL-GO 3D printed composites is composed of $51 \times 25$ pixels with 1015 spectral bands between 996 cm$^{-1}$ and 2111 cm$^{-1}$. To reduce the dimensionality of and to project the data into a linear subspace, the PCL-GO map has been processed through an AANN. Following an iterative process varying the number of nodes in the hidden layers, it has been found that the best topology, having the lowest training MSE presented a 1015-500-25-500-1015 topology. This means that the AANN has the 1015 bands as the input, 500 nodes in the coding/decoding layer, and then 25 nodes in the bottleneck layer. The features obtained through the bottleneck layer have then been used as input to the N-FINDR algorithm considering only two endmembers. Subsequently, the obtained endmembers have been used to determine the fractional abundances in each pixel of the map.

The obtained results have been compared with those collected using the classical linear unmixing approach, without the projection in the feature space. Analyzing Figure 4, showing the reference spectra and the endmembers resulting from the linear approach, it is possible to note that while the first endmember, noted as EM#1, is very similar to the PCL reference spectra, the second endmember, noted as EM#2, presents spectral characteristics more similar to a mixed spectrum rather than pure GO. This can be clear comparing the response of features 18, 21, and 25 reported in Figure 5.

Table 1 reports the SAD values obtained comparing the reference spectra of GO and PCL, projected in the feature space, with the endmembers obtained with both linear and nonlinear approaches, respectively. Referring to the linear approach, the SAD values for EM#2 present almost the same value for the comparison with PCL and GO. This means that using the linear approach is not possible to clearly associate EM#2 to any reference spectra. On the other hand, considering the nonlinear approach, the SAD values for EM#1 is lower when compared to PCL, while EM#2 present the lower value when compared to GO, allowing the association of the two derived endmembers to the two materials.

**Table 1.** SAD values obtained by comparing the reference spectra of GO and PCL, and the endmembers obtained using the linear and nonlinear approaches.

|  | Linear Unmixing Approach | | Nonlinear Unmixing Approach | |
| :---: | :---: | :---: | :---: | :---: |
|  | GO | PCL | GO | PCL |
| EM#1 | 73.54 | 67.92 | 0.0028 | 0.0012 |
| EM#2 | 69.15 | 69.93 | 0.0015 | 0.0033 |

For sake of comparison, following a traditional approach, we also estimated the similarities of detected spectra with a reference spectral library in terms of SAD values. Figure 6 reports the comparisons between the GO and PCL spectra and the most similar spectra in terms of SAD. As can be noted, this approach is possible to identify correctly only PCL, which is the dominant material. Indeed, this approach fails to identify GO mainly because it is present only in mixed solutions.

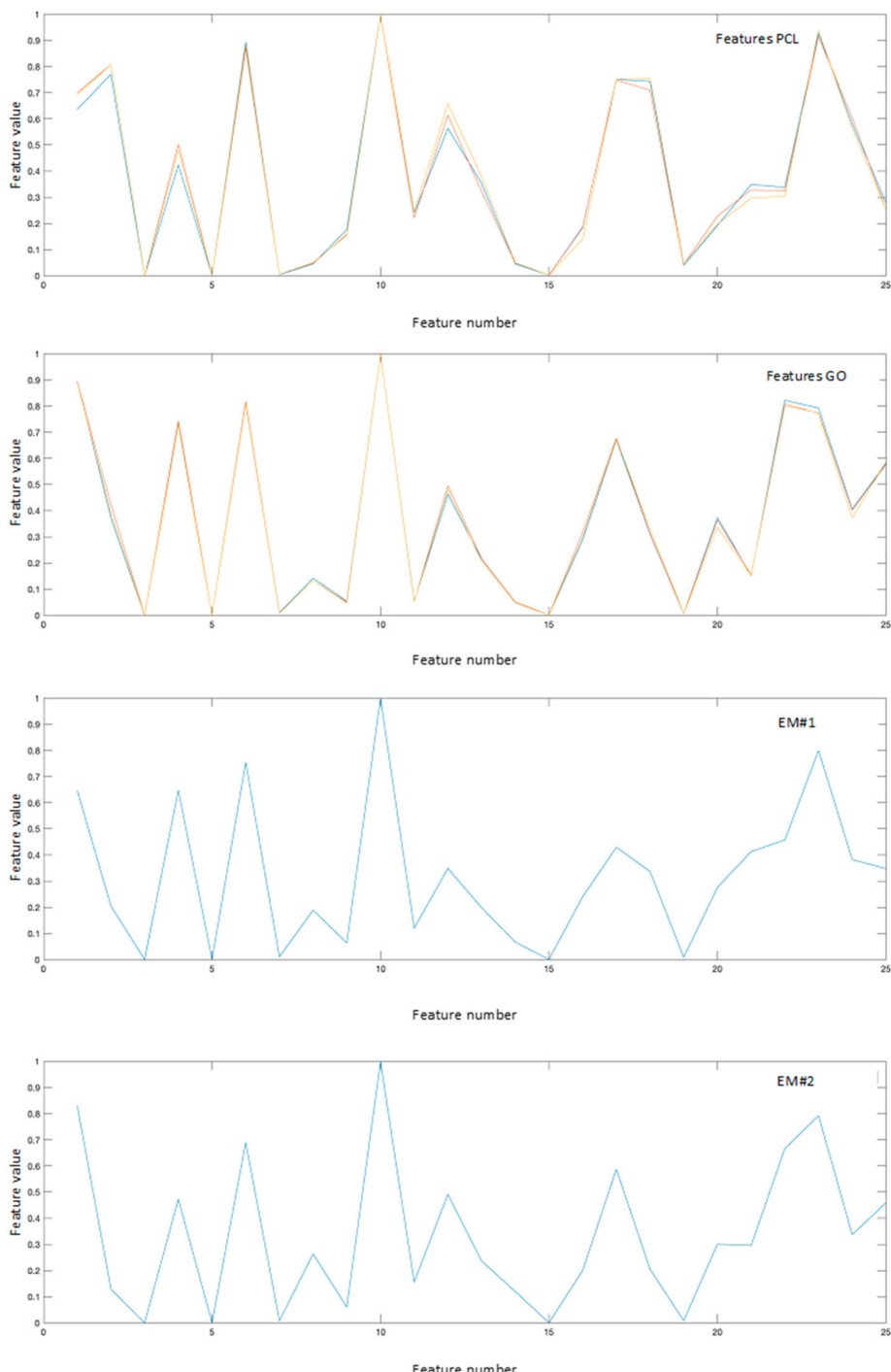

**Figure 5.** Reference features obtained projecting the reference spectra in the feature space, and the endmembers obtained with the proposed nonlinear approach.

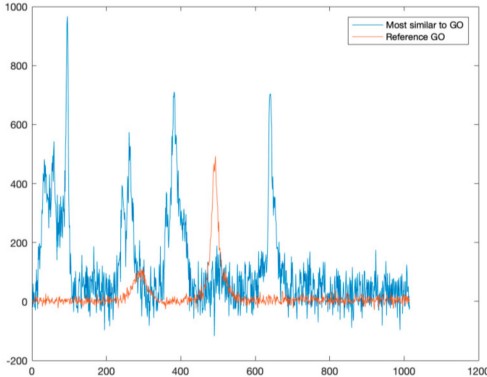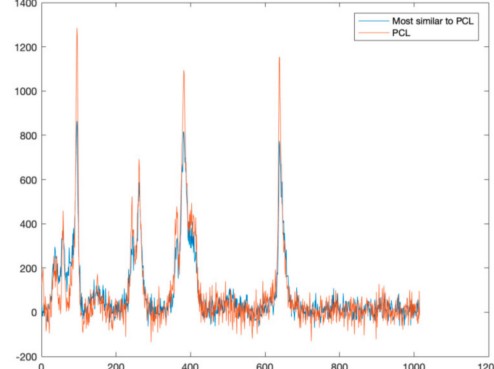

**Figure 6.** Comparisons between reference spectra (GO on the **left**, PCL on the **right**), and the spectra most similar in terms of SAD.

In a second evaluation the reference features, obtained from the projection of the reference spectra in the feature space have been used as endmembers to determine the fractional abundances of each pixel. The RMSE values have been derived by evaluating the errors between the original map and the reconstructed map obtained with Equation (9), where $\Phi_z(i,j)$ and $E_z$ are respectively the endmembers and the fractional abundances obtained using the linear unmixing approach on the original data. Regarding the proposed nonlinear approach, the RMSE values have been calculated using the low dimensionality feature map, obtained with the AANN, and the reconstructed feature map obtained using Equation (9), where $\Phi_z(i,j)$ and $E_z$ are respectively the endmembers and the fractional abundances derived using the linear approach in the feature space.

Table 2 reports the RMSE values obtained after reconstructing the feature and the original data maps.

**Table 2.** RMSE values derived between the original map and the reconstructed one, using linear (left) and nonlinear (right) approaches.

| RMSE | Linear Approach | Nonlinear Approach |
|---|---|---|
| PCL-GO | 2.24 | 0.0096 |
| PLA-GO | 1.012 | 0.0164 |

Figure 7 graphically represents the per-pixel root mean square error (RMSE) obtained after reconstructing the Raman Spectral maps, using the linear approach and the proposed nonlinear approach, respectively. As can be seen, the proposed nonlinear method improves the quality of the map reconstruction.

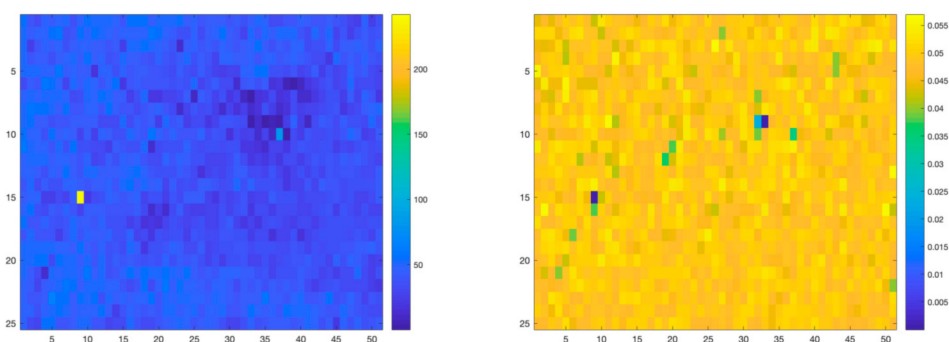

**Figure 7.** RMSE reconstruction errors for linear (**left**) and nonlinear (**right**) unmixing algorithms applied to the PCL-GO structure.

### 4.2. PLA-GO Dataset

A second experiment has been carried out on a 3D structure of PLA including GO. The obtained map is composed of $111 \times 17$ pixels with 1015 spectral bands between 996 cm$^{-1}$ and 2111 cm$^{-1}$. In this case an AANN having a 1015-500-25-500-1015 topology has also been used to project the original data into a reduced dimensionality linear feature space. Differently from the previous experiment, we bypassed the N-FINDR algorithm and instead used reference spectra as endmembers to estimate the fractional abundances in each pixel of the map. The RMSE scores of the obtained results have been compared with those obtained through a linear unmixing approach and reported in Table 2. As can be noted, the map reconstructed using the nonlinear approach presents a lower RMSE if compared with the map reconstructed with the linear approach. This can be also appreciated by analyzing the error maps in Figure 8, where the difference values between the original Raman spectral map and the reconstructed map obtained using the proposed nonlinear approach are lower than the difference values between the original Raman map and the reconstructed map obtained using the linear approach.



**Figure 8.** RMSE reconstruction errors for the linear (**left**) and nonlinear (**right**) unmixing algorithms applied to the PCL-GO structure.

Similarly, to the previous experiment, we applied a SAD-based technique to detect and identify all the different materials present in the sample. The identification is carried out by deriving a similarity index obtained by comparing each pixel of the Raman map with a set of reference spectra based on the SAD value. Figure 9 reports the comparison between the most similar spectra and the reference spectra for PLA and GO, respectively. In this case, while it is possible to correctly identify the PLA, which is the dominant material, this approach does not allow the detection of GO, which is only present in a mixed solution.

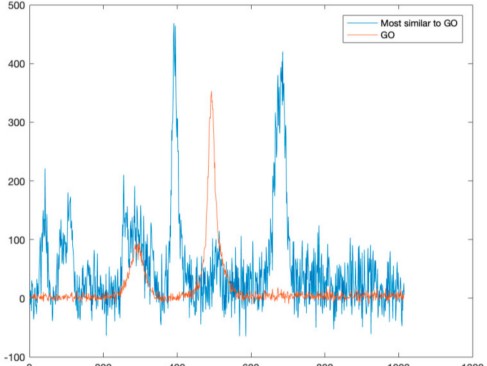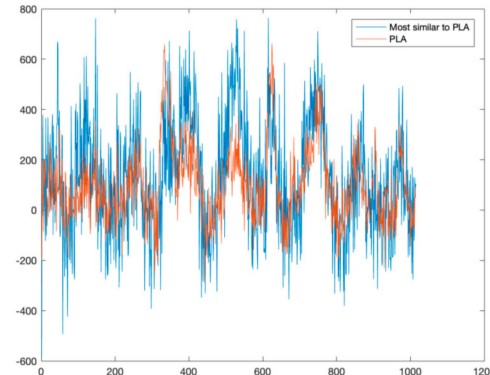

**Figure 9.** Comparisons between reference spectra (GO on the **left**, PLA on the **right**), and the spectra most similar in terms of SAD.

## 5. Conclusions

In this paper, we presented a novel approach to detect and identify different materials in Raman spectra acquired on 3D structures. Traditional methods to identify materials from Raman spectroscopy are based on the comparison of the detected spectra with predefined spectral libraries. When the sample presents mixtures of different materials, traditional methods tend to fail. In this paper, we propose to use the nonlinear generalization of the spectral unmixing approach that allows us to identify different materials even if they are present in a mixed solution. Nonlinear intimate mixtures of the pure spectra of the single materials mostly characterized the detected Raman spectra. Modeling the mixtures with linear approaches may lead to a non-detection of all the endmembers in the data. Nonlinear approaches in general require extremely complex nonlinear models of the interactions between light and matter that are far from a practical realization.

The proposed approach takes advantage of an auto-associative neural network (AANN), a deep autoencoder which allows us to project the original Raman spectra into a low-dimensionality linearized feature space. The use of the AANN allows for the solving of nonlinear correlations in the original data. Thus, it is possible to apply the linear unmixing approach directly in the features space.

The effectiveness of the proposed approach has been tested on two real datasets acquired on 3D printed structures including GO in PCL and PLA matrices, respectively. The obtained results have been compared with those obtained applying the linear approach to the same data and evaluated in terms of Spectral Angle Distance (SAD) and Root Mean Square Error (RMSE). The results demonstrate that the proposed approach overcomes the limitations introduced by linear unmixing and does not require complex modeling of the light-matter interactions.

The full and straightforward implementation of the method presented here can be suitably and readily considered for the evaluation of practical cases in different sectors. Dealing with a hyperspectral camera can monitor processes and materials by remote control, providing a number of benefits: (i) there is no need to handle the samples, this could be particularly interesting for addressing, e.g., safety issues; (ii) sample size is not a limitation, as occurred in this experimentation as samples had to be located under a microscope, and (iii) the collected data can be directly processed by the unmixing algorithm here proposed.

**Author Contributions:** Conceptualization, G.L. and C.D.G.; methodology, G.L.; validation, G.L. and C.D.G.; supervision, J.C. All authors have read and agreed to the published version of the manuscript.

**Funding:** This research received no external funding.

**Conflicts of Interest:** The authors declare no conflict of interest.

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
