# Peer review of "Non-Linear Spectral Unmixing for the Estimation of the Distribution of Graphene Oxide Deposition on 3D Printed Composites"

_applsci, doi:10.3390/app10217792_

Round 1

Reviewer 1 Report

I feel the quality and novelty is not sufficient to be published in Applied Sciences. I am sorry that I have to reject this draft. Welcome to submit again if the below issues can be addressed:

  1. Texts in Figure 1 have strange symbols;
  2. Figure 2 is not necessary;
  3. A figure on your experimental setting is needed, at least the 3D printer and with the printed structures.
  4. There is hyperspectral imaging introduction in the abstract and introduction parts, but I did not find the data or photos in the draft.
  5. The experimental data is so simple. Can you add more experimental data on structure or applications of your real samples.
  6. A concise description of the novelty of the work needs to be addressed.

Author Response

Reviewer 1

Texts in Figure 1 have strange symbols.

The strange symbols were produced during the transformation of the manuscript from docx to Pdf. We hope this won’t happen in future.

Figure 2 is not necessary.

Figure 2 has been removed.

A figure on your experimental setting is needed, at least the 3D printer and with the printed structures.

The Authors apologize for the inconvenience and the main text has been modified, reporting the sample size and the related CAD model (shown in the following figure).

There is hyperspectral imaging introduction in the abstract and introduction parts, but I did not find the data or photos in the draft.

An RGB image showing the superimposition of the Raman spectral maps obtained over the two samples is reported in Fig. 3.

The experimental data is so simple. Can you add more experimental data on structure or applications of your real samples.

Aim of the proposed study is to highlight the potential of an alternative investigational technique for materials characterization. The deposition of GO on 3D printed materials is usually carried out because of the improvements introduced by a second phase in terms of e.g. structural properties, surface characteristics, and chemical responses of materials. However, the effectiveness of these improvements depends on the uniform and homogeneous deposition of GO over the whole surface of the material. Thus, an instrument able to inspect the uniformity of deposition of GO over 3D printed materials may be important for the characterization of the composites in many industrial cases. Moreover, the effectiveness of the proposed approach in identifying different endmembers in multivariate data, such as the Raman spectra, has been already proved in [1], where the nonlinear spectral unmixing has been applied to EO hyperspectral data. In order to extend to other different application fields the presented methodology, we preferred to test our hypothesis on PLA-GO and PCL-GO composites as a preliminary study.

[1] G. Licciardi, P. Sellitto, A. Piscini and J. Chanussot, “Non-Linear Spectral Unmixing for the Characterisation of Volcanic Surface Deposit and Airborne Plumes from Remote Sensing Imagery”, Geosciences, 2017, 7, 46

A concise description of the novelty of the work needs to be addressed.

The authors thank the Reviewer for this suggestion, which has been added at the end of the Introduction section, as follows:

“The here presented study is aimed to highlight the potential of an alternative investigational technique for materials characterization. As a case study, commercial polycaprolactone and polylactic acid filaments were modified with GO to 3D print testing composites, in order to implement Raman findings for the spectral unmixing analysis. Raman mapping is generally carried out fixing a specific wavenumber representative of the material to be detected; however, this might lead to an uneven intensity representation related to possible peak shifts which can bias the acquired results. Differently, hyperspectral imaging needs a minimal set of input data, i.e. the spectral signatures of the neat materials, to directly identify the searched nanomaterial, which allows to limit the number of the experimental constraints to be set and thus perform a reliable material assessment.”

Reviewer 2 Report

The authors discuss the performances of a custom-made algorithm for hyperspectral image analysis of the Raman image of a 3D printed polymer matrix composite. The algorithm aims at identifying the graphene oxide distribution on samples surface starting from the spatial and spectral information.

The paper explains the working principle of the custom-made algorithm, and verify its performances by application to the Raman image. Generally, the manuscript is very well-written and organized. I am not very much into the development of numerical algorithms for the analysis of hyperspectral datasets; hence I will not go into the mathematical details of the approach. I will instead provide general comments and inquiries on the development and application of the method, and on the data analysis.

1. First of all: in section 3.2 the authors mention and describe the N-FINDR method. However: what is explained in the section is a description of an existing method, or the explanation of a variant elaborated by the authors? This aspect should be better clarified.

2. The authors describe the preparation of the sample, but still I do not get its topology. Since it is 3D printed, what is the size of the sample in the z coordinate? How is it arranged?

3. The authors mention that the scanned area is “randomly selected”. What do they mean? Does this randomness play a role in the analysis? Or they simply mean that they selected one generic area of the sample?

4. Is the colorbar scale right, in Figs. 7 and 8? In both cases, left and right panels differ by a ratio larger than 10^4. This seems in contrast with the numbers reported in table 2; maybe Table 2 and maps of figs 7 and 8 ddeal with different aspects? This should be better explained. In addition, a label should be associated to the colorbars.

5. The method should identify the endmembers and ultimately the species of the sample: did the authors obtain this goal? Could they precisely identify exactly the species at each point of the sample? Did they cross-check the results by other experimental methods?

6. Some other minor comments:

- line 88: I think it should read “... in the pixel (i,j).” instead of “... in the pixel X(i,j).”

- Labels are missing for both axes in Fig.6. Font size should be also increased.

I believe that the paper should be published, provided that the authors address these comments.

Author Response

Reviewer 2

  1. In section 3.2 the authors mention and describe the N-FINDR method. However: what is explained in the section is a description of an existing method, or the explanation of a variant elaborated by the authors? This aspect should be better clarified.

Indeed, the N-FINDR method has been introduced in [2] and applied to different EO hyperspectral data. In this manuscript we use this same approach applied to Raman spectra. A line explaining this has been added to the text.

[2] Plaza, A.; Chang, C.I. An improved N-FINDR algorithm in implementation. Proc. SPIE 5806, Algorithms and Technologies for Multispectral, Hyperspectral, and Ultraspectral Imagery XI 2005.

  1. The authors describe the preparation of the sample, but still I do not get its topology. Since it is 3D printed, what is the size of the sample in the z coordinate? How is it arranged?

The Authors apologize for this inconvenience. Sample size was 8 x 8 x 3.2 mm3, fabricated according to the CAD model shown in the figure below.

  1. The authors mention that the scanned area is “randomly selected”. What do they mean? Does this randomness play a role in the analysis? Or they simply mean that they selected one generic area of the sample?

The Authors thank the Reviewer for this comment which give us the possibility to better explain the implemented experimental approach. The random selection of the areas simply means that the investigated samples showed a homogenous distribution of the nanofiller and this was actually verified avoiding to focus on specific surface regions. In other words, generic areas of the samples were selected as the regions to be analysed are not dependent on a particular position.

  1. Is the colorbar scale right, in Figs. 7 and 8? In both cases, left and right panels differ by a ratio larger than 10^4. This seems in contrast with the numbers reported in table 2; maybe Table 2 and maps of figs 7 and 8 deal with different aspects? This should be better explained. In addition, a label should be associated to the colorbars.

We double checked the data and indeed we found that the RMSE values for the linear method were in another scale and needed to be multiplied x100. The scales on figures 7 and 8 are correct since they take into account the strong errors values in some spots.

  1. The method should identify the endmembers and ultimately the species of the sample: did the authors obtain this goal? Could they precisely identify exactly the species at each point of the sample? Did they cross-check the results by other experimental methods?

In the first experiment we applied the N-FINDR algorithm to the reduced dimensionality feature data in order to detect exactly 2 endmembers and subsequently the Sunsal algorithms was used to estimate endmember’s abundances. Table 1 of the manuscript reports the SAD values obtained comparing the reference spectra of GO and PCL, projected in the feature space, with the endmembers obtained with both linear and nonlinear approaches, respectively. Referring to the linear approach, the SAD values for EM2 present almost the same value for the comparison with PCL and GO. This means that using the linear approach is not possible to clearly associate EM#2 to any reference spectra. On the other hand, considering the nonlinear approach, the SAD values for EM1 is lower when compared to PCL, while EM2 present the lower value when compared to GO, allowing the association of the two derived endmembers to the two materials. We added a paragraph to better explain this point.  

  1. Some other minor comments:

- line 88: I think it should read “... in the pixel (i,j).” instead of “... in the pixel X(i,j).”

Indeed the term X(i,j) is correct since it indicates the pixel vector at position (i,j). Maybe the text is misleading, so we changed “... in the pixel X(i,j).” into “... in the pixel vector.”

- Labels are missing for both axes in Fig.6. Font size should be also increased.

Fixed

Reviewer 3 Report

The work proposed an interesting pipeline for HSI unmixing. The work is nicely presented, however, there are a few minor suggestions which I believe need to be addressed before the final recommendation of your work. First, the quality of the figures needs to improve, second, the writing of the paper need to be proofread by a native speaker, there are several minor grammatical issues persist in the paper, thrid, there are many other dimensional reduction and feature learning methods exists, which I believe need to be discussed (e.g., "A New Statistical Approach for Band Clustering and Band Selection Using K-Means Clustering", "Hyperspectral Unmixing With Spectral Variability Using Adaptive Bundles and Double Sparsity", "Metric similarity regularizer to enhance pixel similarity performance for hyperspectral unmixing", "A Supervised Method for Nonlinear Hyperspectral Unmixing", etc,) in the paper either in introduction section or make a new section of literature survey. Good Luck!

Author Response

Reviewer 3

The quality of the figures needs to improve

According to the Reviewer’s suggestion, the quality of the figures has been improved.

The writing of the paper needs to be proofread by a native speaker, there are several minor grammatical issues persist in the paper.

The Authors thank the Reviewer for this comment and the manuscript has been critically revised and corrected.

There are many other dimensional reduction and feature learning methods exists, which I believe need to be discussed (e.g., "A New Statistical Approach for Band Clustering and Band Selection Using K-Means Clustering", "Hyperspectral Unmixing With Spectral Variability Using Adaptive Bundles and Double Sparsity", "Metric similarity regularizer to enhance pixel similarity performance for hyperspectral unmixing", "A Supervised Method for Nonlinear Hyperspectral Unmixing", etc,) in the paper either in introduction section or make a new section of literature survey.

We agree with the reviewer that there exist several dimensionality reduction methods that apply to hyperspectral data. However, the aim of this manuscript is to demonstrate that the proposed method, which was positively applied to remote sensing hyperspectral data, can effectively be used for spectral unmixing of Raman spectra. A detailed description of different techniques for dimensionality reduction compared to the proposed approach are reported in [3], while a complete overview of the different unmixing techniques present in the literature are reported in [4]. References to these papers have been inserted in the text of the manuscript.

[3] Giorgio Licciardi & Jocelyn Chanussot (2018) Spectral transformation based on nonlinear principal component analysis for dimensionality reduction of hyperspectral images, European Journal of Remote Sensing, 51:1, 375-390

[4] Bioucas-Dias, J. M., Plaza, A., Dobigeon, N., Parente, M., Du, Q., Gader, P., & Chanussot, J. (2012). Hyperspectral unmixing overview: Geometrical, statistical, and sparse regression-based approaches. IEEE journal of selected topics in applied earth observations and remote sensing, 5(2), 354-379.

Round 2

Reviewer 1 Report

  1. Figure 1 and 4 still have the strange symbols.
  2. No full stop at line 136.
  3. Characteristic of the specimens samples needs to be given, including the thickness, roughness and so on (line 216). There is not enough a figure for the real sample or experimental setup.
  4. The results which can be generated by this method+
  5. do not show clearly. The experimental data is so simple and cannot reflect in abstract or conclusion. I cannot see the novelties of using this method from the results. Please add more experimental data before next submission.
  6. Even though you have addressed some novelties to the last reviewers, it is still difficult to follow. The sections for featured application and specific application are suggested.

Author Response

Non-linear spectral unmixing for the estimation of the distribution of graphene oxide deposition on 3D printed composites

applsci-933671

Round 2

The Authors thank the Reviewer once again for reading and commenting our manuscript.

Please find the point-by-point replies in the following.

  1. Figure 1 and 4 still have the strange symbols.

Considering that this comment has been reproposed, the Authors did not correctly receive this suggestion and a further clarification could be helpful, for instance listing the “strange symbols”. We may suppose that the strange symbols refer to the letters shown in the figures and, in order to improve the readability, a legend was added to the figure captions. However, in the PDF file we submitted, neither in the doc file, these strange symbols are not present. We suppose these are the result of the format conversion in the journal website.

  1. No full stop at line 136.

Fixed.

  1. Characteristic of the specimens samples needs to be given, including the thickness, roughness and so on (line 216). There is not enough a figure for the real sample or experimental setup.

Specimens characteristics were not here fully reported, being not the main argument of the study. In order to provide specific information on this issue the main text was modified, inviting the Reader to refer to Ref. 1 for a detailed analysis of the resulting scaffolds. A figure of the 3D printed scaffolds has been added to Section 4.

  1. The results which can be generated by this method do not show clearly. The experimental data is so simple and cannot reflect in abstract or conclusion. I cannot see the novelties of using this method from the results. Please add more experimental data before next submission.

Traditional methods to identify materials from Raman spectroscopy are based on the comparison of the detected spectra with predefined spectral libraries. When the sample presents mixtures of different materials, traditional methods tends to fail. In this paper we propose a novel method based on the nonlinear generalization of the spectral unmixing approach that permits to identify different materials even if they are present in a mixed solution. Linear spectral unmixing techniques are not new in the literature and have demonstrated to effectively separate spectral contributions that are linearly mixed. However, linear approaches tend to fail when the mixture is nonlinear. Thus, the novelty introduced with the proposed solution resides in the ability of the technique to deal with the nonlinear interactions between light and matters that characterize the Raman spectra. To further emphasize this concept we added a comparison with a traditional method that compares the similarity between the detected and reference spectra in terms of SAD. Moreover, a more clear explanation has been also added in the Introduction and in the Conclusions sections.

  1. Even though you have addressed some novelties to the last reviewers, it is still difficult to follow. The sections for featured application and specific application are suggested.

The proposed method is derived from the analysis generally used for Earth observation and it implies that is possible to share all the advantages of this kind of approach for several practical cases in different sectors. For instance, it is possible to carry out a non-contact analysis, differently from the one here described in which samples had to be physically handled to carry out the Raman assessment, considered a comparative test case. To overcame this limitation, a hyperspectral camera can allow to monitor processes and materials in remote control, also dealing with large-scale samples, to collect the necessary input to be processed by the unmixing algorithm here discussed.

These considerations and suggestions towards possible applications have been added to the Section Conlusions.

Round 3

Reviewer 1 Report

I did not find a figure of the 3D printed scaffolds been added to Section 4. More experimental data with a specific application is needed for publication. So I am sorry, I have to decline this version for publication.

Author Response

The Authors thank the Reviewer once again for reading and commenting our manuscript.

Please find the point-by-point replies in the following.

I did not find a figure of the 3D printed scaffolds been added to Section 4.

An error occurred in the previous version of the manuscript. Now the figure of the 3D scaffolds has been correctly incuded.

More experimental data with a specific application is needed for publication.

We added more experimental data with a comparison with a traditional method for the identification of materials in samples.